# Preparation and Characterization of Nano-Fe_3_O_4_ and Its Application for C18-Functionalized Magnetic Nanomaterials Used as Chromatographic Packing Materials

**DOI:** 10.3390/nano13061111

**Published:** 2023-03-20

**Authors:** Wen-Xin Liu, Wei-Na Zhou, Shuang Song, Yong-Gang Zhao, Yin Lu

**Affiliations:** 1College of Environment, Zhejiang University of Technology, Hangzhou 310014, China; lwx187221@163.com (W.-X.L.); ss@zjut.edu.cn (S.S.); 2College of Biological and Environmental Engineering, Zhejiang Shuren University, Hangzhou 310015, China

**Keywords:** solvothermal method, chromatographic packing materials, magnetic Fe_3_O_4_ particles, C18-functionalized magnetic nanomaterials

## Abstract

A new type of magnetic nanomaterial with Fe_3_O_4_ as the core and organic polymer as the shell was synthesized by seed emulsion polymerization. This material not only overcomes the problem of insufficient mechanical strength of the organic polymer, it also solves the problem that Fe_3_O_4_ is prone to oxidation and agglomeration. In order to make the particle size of Fe_3_O_4_ meet the requirement of the seed, the solvothermal method was used to prepare Fe_3_O_4_. The effects of the reaction time, amount of solvent, pH value, and polyethylene glycol (PEG) on the particle size of Fe_3_O_4_ were investigated. In addition, in order to accelerate the reaction rate, the feasibility of preparing Fe_3_O_4_ by microwave was studied. The results showed that under the optimum conditions, the particle size of Fe_3_O_4_ could reach 400 nm and had good magnetic properties. After three stages of oleic acid coating, seed emulsion polymerization, and C18 modification, the obtained C18-functionalized magnetic nanomaterials were used for the preparation of the chromatographic column. Under optimal conditions, stepwise elution significantly shortened the elution time of sulfamethyldiazine, sulfamethazine, sulfamethoxypyridazine, and sulfamethoxazole while still achieving a baseline separation.

## 1. Introduction

At present, chromatographic packing materials are mainly silicon-based materials because the silicon-based materials have a strong chemical stability, and the surface is rich in silicon hydroxyl groups [1,2,3]. Therefore, it can be modified according to the needs of separation and analysis. In addition, silicon-based materials also have high mechanical strength and stability. All these ensure that the chromatographic column prepared by silicon-based materials has the advantages of wide range of application, good reproducibility, high stability, high column efficiency, low resistance, and so on [4,5,6,7]. However, the preparation process of silicon-based materials is complex, and the manufacturing cost is high. In order to find a substitute with a simple preparation process and low manufacturing cost, organic polymer has become an optional choice. Because it is simple to prepare an organic polymer, the polymerization can be initiated by evenly mixing and degassing the polymerization solution composed of a functional monomer, a cross-linking agent, an initiator, and a pore-forming agent under the condition of light or heating. In the preparation process, the pore size and the particle size can be controlled by adjusting the ratio of the pore-inducing agent and the functional monomer. However, the mechanical strength of organic polymers is low, which is the main factor hindering their application in chromatographic packing materials.

In order to solve this problem, it is necessary to combine organic polymers with other materials. The Fe_3_O_4_ nano-material is one of the materials that can be compounded because it contains hydroxyl groups and can be chemically modified. Additionally, Fe_3_O_4_ has high magnetization, superparamagnetism, excellent physical properties, stability [8,9,10], good biocompatibility, and unique electromagnetic properties [11,12,13]. Owing to its above properties, Fe_3_O_4_ is used in the fields of separation [14,15,16], protein immobilization [17], catalysis [18,19,20], medicine [21,22], environment [23,24], and so on. However, Fe_3_O_4_ nanoparticles have a huge defect, which is that it is easy to agglomerate and oxidize because of its high energy state [25,26]. Therefore, in order to prevent the agglomeration and oxidation of Fe_3_O_4_, it is necessary to form the structure of Fe_3_O_4_ encapsulated by an organic polymer when compounded with an organic polymer. This structure not only ensures the sufficient mechanical strength of the organic polymer, it also ensures the oxidation resistance and dispersion of Fe_3_O_4_.

In order to better realize the structure of the organic polymer-coated Fe_3_O_4_, suspension polymerization and seed emulsion polymerization were investigated [27]. The preparation process of suspension polymerization is simple, but the particle size uniformity of the prepared nanoparticles is poor. The range of particle size distribution and the position of Fe_3_O_4_ are both difficult to control. Although the preparation process of seed emulsion polymerization is relatively complex, the particle size is more uniform, and the control over the particle size is simpler. The most important thing is that Fe_3_O_4_ can be used as the seed of organic polymerization, which can accurately control the position of Fe_3_O_4_ in the microspheres. However, only when the particle size of ferric oxide is large enough can it be used as seed for seed emulsion polymerization.

Generally speaking, the preparation methods of Fe_3_O_4_ nanoparticles include coprecipitation [28,29], pyrolysis [30], microemulsion [31], the solvothermal method [32,33,34,35], the modified polyol method [36], the electrochemically assisted synthesis method [37], the sonochemical method [38], and so on. Of all these methods, coprecipitation is the first choice for most researchers because the reaction can be carried out with water as a solvent under mild conditions. However, the shape of the finished product is difficult to control, and the particle size is generally smaller than 20 nm. Pyrolysis is considered to be the best way to obtain nanomaterials with controllable sizes and morphologies. However, the precursors of the Fe_3_O_4_ are organometallic compounds, such as Fe(CO)_5_, metal copper ferrate, and metal acetoacetate, which are toxic. In addition, the pyrolysis of organometallic compounds often requires a very high temperature, which is a great challenge for many populations. The microemulsion method can be used to synthesize nanoparticles with specific morphologies and uniform sizes. Unfortunately, this method consumes a lot of organic solvents and requires organic surfactants to control the size and shape of the product. The Fe_3_O_4_ prepared by the polyol method has hydrophilic polyol groups on its surface and can be dispersed in water or other polar solvents with high magnetic properties, but the cost of this method is high, and the residue can easily cause environmental pollution. The sonochemical method has the advantages of simple operation, easy control, high efficiency, narrow particle size distribution, and no need for an additional initiator. However, the nanoparticles prepared by the sonochemical method are prone to agglomeration and other problems. Compared with the above methods, the solvothermal method has the advantages of high yield and controllable particle size. Additionally, almost all materials can be dissolved at higher pressure and higher temperature, which greatly increases the solubility of solids, resulting in the synthesis of high-quality, nanostructured materials. To sum up, the solvothermal method has become the best choice to obtain Fe_3_O_4_ particles with larger particle sizes.

In this study, Fe_3_O_4_ prepared by the solvothermal method was used for seed emulsion polymerization to prepare a new type of composite material, with magnetic material for the core and organic polymer for the shell (Figure 1). During the preparation of Fe_3_O_4_, the reaction time, pH, the amount of solvent, and polyethylene glycol were optimized. Then, the optimized Fe_3_O_4_ was coated with oleic acid using seeded emulsion polymerization and C18 functionalization. The magnetic nanocomposites were used as the chromatographic packing materials to prepare the reversed-phase chromatographic column. Finally, the obtained column was applied in the analysis of four sulfonamides.

## 2. Experimental Section

### 2.1. Reagents and Materials

Ferric chloride hexahydrate (FeCl_3_·6H_2_O), sodium acetate (NaAc), oleic acid (OA), ethylene glycol (EG), polyethylene glycol (PEG), polyvinyl pyrrolidone (PVP), styrene (ST), divinylbenzene (DVB), 2,2′-Azobis(2-methylpropionitrile) (AIBN), sodium dodecyl sulfate (SDS), dibutyl phthalate (DBP), polyvinyl alcohol (PVA), benzoyl peroxide (BPO), N, N-dimethyloctadecylamine, and glycidyl methacrylate (GMA) of analytical grade were purchased from Sinopharm Chemical Reagent Co., Ltd. (Shanghai, China) (Appendix A). Methanol of HPLC grade and the standard solution of sulfamethyldiazine, sulfamethazine, sulfamethoxypyridazine, and sulfamethoxazole (1000 mg/L) were purchased from Aladdin.

### 2.2. Equipment

The magnetic and morphological characteristics of Fe_3_O_4_ were carried out by using an S4800 field-emission scanning electron microscope (SEM, Hitachi, Tokyo, Japan), an Ultima IV polycrystal (powder) X-ray diffractometer (XRD, Tokyo, Japan), and a Lake Shore 7404 vibrating sample magnetometer (VSM, Westerville, USA). The specific surface area and pore size of the adsorbent were measured and calculated by Brunauer–Emmett–Teller (BET) and Barrett–Joyner–Halenda (BJH) (Micromeritics, Norcross, USA).

### 2.3. Preparation of Magnetic Fe_3_O_4_ Particles

The preparation of Fe_3_O_4_ nanoparticles by the solvothermal method was as follows: 2.0 g of FeCl_3_·6H_2_O was added to a certain amount of EG solution, and a certain amount of PEG was added to the solution, which was completely dissolved by ultrasound. Then, 6.0 g of NaAc was added to the mixture and stirred evenly by ultrasound. It was transferred and sealed in a 100 mL Teflon-lined stainless-steel autoclave and reacted at 200 °C. After 8 h of reaction, it was cooled to room temperature and washed several times in sequence with deionized water and anhydrous ethanol. Finally, the product was collected using a magnet and baked in an oven at 70 °C for 12 h to obtain Fe_3_O_4_ nanoparticles. In order to explore the mechanism of Fe_3_O_4_ synthesis and the factors affecting the particle size, the effects of NaOH concentration (A), reaction time (B), usage of PEG (C), solvent volume (D), and microwave preparation time (E) were set (Table 1).

### 2.4. Preparation of C18-Functionalized Magnetic Nanomaterials

#### 2.4.1. Preparation of Magnetic PS-DVB-GMA Materials

In this work, magnetic PS-DVB-GMA materials were prepared by seeded emulsion polymerization as follows:(1)A total of 1.0 g of Fe_3_O_4_ was mixed with 200 mL of anhydrous ethanol. Then, 5.0 mL of OA was slowly dripped at 80 °C. After that, the reaction lasted for 1.0 h.(2)An amount of 1.0 g OA-Fe_3_O_4_ was mixed with 100 mL of 95% ethanol aqueous solution. Then, certain amounts of PVP, ST, and AIBN were added to it, mixed evenly, and transferred to the three-neck flask of 250 mL. The reaction was carried out in a nitrogen atmosphere at 70 °C for 24 h.(3)A total of 4 mL of DBP were dispersed in 50 mL of SDS (0.2%, *m*/*v*) solution and emulsified in the ice bath for 1 h. Then, it was mixed with the prepared seeds and transferred to a 500 mL three-neck flask and stirred at 30 °C for 24 h.(4)A certain amount of SDS and BPO were added to the 250 mL PVA. Toluene, DVB, and GMA were added to the solution at a ratio of 3:1:1. Then, it was emulsified in the ice bath for 2 h. After that, it was poured into the three-neck flask in step 2 and stirred continuously at 30 °C for 24 h.(5)After passing nitrogen for 30 min, the temperature rose to 70 °C for the 24 h reaction. After the reaction, the particles were washed with hot water and anhydrous ethanol several times and then dried at 70 °C.

#### 2.4.2. C18-Functionalized Magnetic Nanomaterials

The 4.0 g magnetic PS-DVB-GMA materials were uniformly mixed with 100 mL of methanol and transferred to the 250 mL three-neck flask. Under the conditions of 80 °C and 500 rpm, 10 mL of N, N-dimethyloctadecylamine was slowly added. Then, there was a reflux reaction for 8 h. At the end of the reaction, the nanoparticles were washed with methanol and anhydrous ethanol several times and then dried. Finally, C18-functionalized magnetic nanomaterials were obtained. 

The as-prepared C18-functionalized magnetic nanomaterials were ultrasonically suspended in 50.0 mL of cyclohexanol, and then the slurry-packed sorbent was transferred to stainless-steel column tubes (4.6 × 250 mm) at 40 MPa. The packed column was separated and plugged at the column inlets. After being eluted with 200 mL of methyl alcohol, the C18 column was successfully prepared and expected to be applied in the separation of sulfamethyldiazine, sulfamethazine, sulfamethoxypyridazine, and sulfamethoxazole.

### 2.5. Standard Preparation 

The standard solutions of single-standard sulfamethyldiazine, sulfamethazine, sulfamethoxypyridazine, and sulfamethoxazole, with 1 mL concentrations of 1000 mg/L, were taken in a 10 mL volumetric flask, had the volumes fixed with methanol, and were prepared into a standard solution with a concentration of 100 μg/mL. Then, five groups of mixed standard solutions with different concentrations of 0.1 μg/mL, 0.5 μg/mL, 1.0 μg/mL, 5.0 μg/mL, and 10 μg/mL were fixed with methanol.

### 2.6. Chromatographic Conditions

The chromatographic separation was performed on the prepared C18 column (4.6 × 250 mm) by using 0.1% formic acid (*v*/*v*) in water as the eluent (A) and methanol as the eluent (B) in the mobile phase. Detector wavelengths were 270 nm, and the temperature of the column thermostat was 33 °C. The injection volume was 10 μL. The separation was accomplished at a constant flow of 1.0 mL/min. Elution gradients: 0.0–6.0 min, 17–20%B; 6.0–11.0 min, 20–25%B; 11.0–35.0 min, 25–75%B; 35.0–40.0 min, 75–17%B; 40.0–45.0 min, 17%B.

## 3. Results and Discussion

### 3.1. Analysis of Synthesis Mechanism

In the reaction system, the role of EG was to provide a mild environment for the reaction. While at a high temperature, trivalent iron was reduced to divalent iron as a reductant; sodium acetate was a strong alkali and a weak acid salt, and acetate reacted with crystalline water in ferric chloride to hydrolyze and produce OH^−^, forming a weak alkaline environment. Then, Fe^3+^ and Fe^2+^ reacted with OH^−^ under high temperature and high pressure to form Fe_3_O_4_ nanoparticles. The reaction equations involved in the experimental system were as follows:CH_3_COO^−^ + H_2_O → CH_3_COOH + OH^−^(1)
Fe^2+^ + 2Fe^3+^ + 8OH^−^ → Fe_3_O_4_ + 4H_2_O(2)

### 3.2. Effect of the Usage Amount of NaOH on the Product

The effect of the usage amount of NaOH on the particle size of the product was investigated deeply, and the results are shown in Figure 2. When the usage amount of NaOH was 0.4 g, the particle size of the magnetic microspheres was the smallest (only about 20 nm), while the particle size of the magnetic microspheres was about 200 nm without using NaOH. The magnetic microsphere was formed by the agglomeration of many small particles (about 20 nm). As shown in Figure 3, the crystalline phases and chemical structures of the prepared particles were determined by XRD. In the XRD diagram, the main peaks of synthesized magnetic microspheres were located at 30.17°, 35.56°, 43.19°, 53.6°, 57.11°, and 62.69°, in line with previously reported works about Fe_3_O_4_ [20]. Therefore, it can be concluded that the prepared particles were magnetic Fe_3_O_4_.

The formation process of the particles from the solution went through the process of nucleation, growth, agglomeration, and so on. The morphology and size of the particles mainly depended on the competition between the nucleation and the growth rate when the particles were precipitated from the solution. When the pH value was high, the metal ions nucleated separately under the alkaline environment. Then, the concentration of metal ions in the solution decreased rapidly, and the particles grew slowly, that is to say, the nucleation rate was greater than the growth rate, which was beneficial to the formation of small particles. When the pH value decreased, there were fewer OH^−^ in the solution; some of the metal ions combined with these OH^−^ to form a nucleus, and the other part was attached to the surface of the crystal nucleus, causing the particles to grow up. At this time, the growth rate was dominant and tended to the formation of large particles. Summing up all of the above analysis results, it was concluded that the phase transformation pH value was an important factor affecting the purity and size of the synthesized nano-Fe_3_O_4_. In order to obtain nanometer particles of three-body tetroxide with a larger particle size, there was no need to add alkaline substances in the reaction solvent, and the pH of the reaction solvent should be controlled in the range of 8–10.

### 3.3. Effect of Reaction Time on Products

Time was one of the important factors in the reaction process. There were four time points: 6 h, 8 h, 10 h, and 12 h. As shown in Figure 4a, it can be seen that when the reaction time was 6 h, the surfaces of the particles were very rough, and there was adhesion between the balls; it was obvious that a large ball was formed by the polymerization of many small balls. This was due to insufficient reaction time. In Figure 4b, the surfaces of the microspheres were much smoother, and the morphology was stable, indicating that the crystal had been fully grown at this time. In Figure 4c, we can see that there were dents or even cracks on the surface of some microspheres, indicating that the crystal had been overgrown at this time, destroying the crystal structure of the microspheres. Figure 4d also confirms the above statement; the crystal had been overgrown in 12 h, and the crystal had seriously collapsed, resulting in crystal splitting and an unstable morphology. From this, it can be concluded that the best reaction time was 8 h.

### 3.4. Effect of the Amount of PEG on the Product

PEG was discussed in this lab. Because PEG can be used as a surfactant, the addition of PEG could improve the dispersion of nanoparticles. As can be seen from Figure 5, the particle size was obviously much more uniform. In addition, statistics were carried out using the software Nano Measurer, and 150 balls were randomly selected from each picture for statistics. The results are shown in Figure 6. The particle size distribution, without adding PEG, was relatively dispersed, while the particle size distribution, with PEG, was concentrated, and the proportion of particle size in the range of 300–450 nm was 76%. It can be concluded that PEG had a great effect on the particle size of Fe_3_O_4_, and the addition of PEG improved the dispersion of nanoparticles in the reaction system, which made the polymerized Fe_3_O_4_ nanoparticles more uniform.

### 3.5. The Mechanism of the Effect of Reaction Solvent Dosage on the Product

The polymerization of nanoparticles was affected by pressure. The pressure of the reaction environment was affected by the amount of solvent. There were five solvent volumes: 40 mL, 50 mL, 60 mL, 70 mL, and 80 mL. A total of 150 microspheres were randomly selected for statistics, as shown in Figure 7, and Figure 8 was obtained. As shown in Figure 7 and Figure 8, in the range of 40 mL–80 mL, the particle size increased with the increase in the amount of solution, which was because the increase in the amount of solution led to the increase in pressure in the container, which made the particles polymerize more easily. When the solvent dosage was 60 mL, the distribution of particle size was the most concentrated, and the proportion of microspheres with particle sizes of 300–450 nm reached 79%. When the solution dose was 40 mL, 74% of the microspheres were smaller than 300 nm, and when the solution dose was 50 mL, the microspheres smaller than 300 nm were reduced to 43%. When the solution dose was 70 mL, the microspheres with particle sizes larger than 400 nm increased by 10%, compared with 60 mL; when the solution dose was 80 mL, the microspheres with particle sizes larger than 400 nm increased by 18%, compared with 60 mL, and the proportion of microspheres with particle sizes less than 300 nm increased to 32%. As a result, it can be seen that the amount of solvent had a great effect on the particle size; with the increase in pressure, there was an increase in the amount of solvent, and the particles were more likely to polymerize under higher pressure. However, too much solvent would widen the range of particle size distribution because a large number of Fe ions were polymerized into large spheres, which reduced the concentration of Fe ions in the reaction system, and could not continue to polymerize to form large spheres.

### 3.6. Analysis of Magnetic Properties of Fe_3_O_4_ Nanoparticles 

Based on the reaction time, the pH, and the polyethylene glycol, the optimum synthesis conditions were determined as the 8 h reaction and a pH of 8~10, and a certain amount of PEG was added. Under these conditions, Fe_3_O_4_ with a uniform particle size was prepared, and its magnetic properties were also tested. The hysteresis loop in the greenhouse was shown in Figure 9. It can be seen that the curve was closed and showed a typical “S” shape. The saturation magnetization of the obtained Fe_3_O_4_ nanospheres was 73.125 emu/g, which had high magnetic properties. In addition, the coercive force (Hc) and remanence (Br) of Fe_3_O_4_ nanoparticles, shown in the upper left magnification, were 1966.78 A/m and 3.0502 A·m^2^/kg, respectively. 

Compared with the previously reported work on the preparation of Fe_3_O_4_, the Fe_3_O_4_ prepared in this study had much more advantages in particle size control and magnetic properties. For example, the average particle size of the Fe_3_O_4_ particles prepared by using the chemical precipitation method was 10 nm, and the magnetization saturation intensity was 60 emu/g [39]. Similar results can be obtained by using the microwave hydrothermal method [40]; the Fe_3_O_4_ particle size was about 20 nm, and the magnetization saturation intensity was also 60 emu/g. As a result, it can be concluded that the Fe_3_O_4_ particles obtained by using the solvothermal method possessed much higher magnetization saturation intensities and larger particle sizes.

### 3.7. Preparation of Fe_3_O_4_ by Microwave

In order to shorten the reaction time, the synthesis of Fe_3_O_4_ by microwave was studied. Microwave could make the reaction reach the required temperature faster and heat more evenly. The time of the microwave was controlled under the conditions of 60 mL, adding 1.0 g of PEG, and keeping the temperature at 200 °C. As shown in the Figure 10, when the reaction time was 2 h, the shapes of the particles were unstable, and there was a flaky structure in addition to the sphere, which shows that the reaction was not complete. When the reaction time was 3 h, there was still a small amount of sheet structures, and the particle size distribution was not uniform. When the reaction time was 4 h, the flake structure disappeared, indicating that the reaction was complete, but the particle size distribution was still uneven. In addition, the particles with a reaction time of 4 h were characterized by VSM, and the range of particle size distribution was calculated. As shown in Figure 11, the Fe_3_O_4_ particles had strong magnetic properties, and its magnetization saturation intensity was 62.49 emu/g. A total of 300 Fe_3_O_4_ particles were randomly selected for statistics, as shown in Figure 12a,b was obtained. It can be concluded that the number of microspheres with particle sizes in the range of 60–100nm was the largest, reaching 35%, and the proportion of microspheres with particle sizes larger than 200 nm was 34%. Compared with previous literature, the particle size of Fe_3_O_4_ prepared by Savvidou [41] was only 20 nm. The particle size of Fe_3_O_4_ prepared by Aivazoglou [41] was 9 nm. It can be seen that the microwave method was mainly used in the preparation of small particle size Fe_3_O_4_. This study provided a certain reference significance for the preparation of large-size Fe_3_O_4_ by microwave.

## 4. Application

### 4.1. Performance Characterization of Materials

According to Figure 13, it can be seen that the new magnetic organic composites had a stable shape, the range of the particle size distribution was narrow, and the average particle size was 4 μm. Through the test, this new type of magnetic composite material had a specific surface area of 300.8128 m^2^/g and a magnetic induction intensity of 27.68 emu/g, which could be used as a filling material for a chromatographic column.

### 4.2. Column Efficiency Analysis

The new type of magnetic composite was prepared into a chromatographic column, and its stereoselectivity and hydrophobicity were tested. The result is shown in Figure 14. The graph shown in Figure 14a is the chromatogram of naphthalene, which was used for the hydrophobicity test. The asymmetry was 0.99, and the number of plates was 11,272. The chromatographic column had a good ability to select and recognize the hydrophobic groups of the separated substances. As shown in Figure 14b, the chromatograms of uracil, ortho-terphenyl, and chrysene were used for stereoselectivity testing, and the stereoselectivity factor was 1.822, which showed that the chromatographic column had a good ability to select and recognize the molecular shape of the separated substance.

### 4.3. Sample Analysis

In order to test the effect of the prepared chromatographic column in the analysis of actual compounds, it was applied to the detection of sulfonamides (Figure 15). It can be seen from Table 2 that the linear range of four kinds of sulfonamides was 0.1–10 μg/mL. The linear correlation coefficients were all above 0.9992. Six groups of parallel experiments were conducted to calculate the relative standard deviation (RSD). The intraday and inter-day RSDs of the four sulfonamides were less than 10.0%, respectively. In addition, the comparison of the prepared chromatographic column with the Diamonsil C18 (5 μm, 4.6 × 250 mm) commercial chromatographic column (Appendix A) was studied, and the results showed that the retention times of four sulfonamides were basically consistent. It could be seen that the magnetic material had a great advantage in the preparation of the chromatographic column.

## 5. Conclusions

To sum up, the preparation of Fe_3_O_4_ using the solvothermal method could be controlled by the pH, the reaction time, the amount of solvent, and polyethylene glycol, and the particle size could be adjusted in the range of 20–400 nm. Among them, the best conditions for preparing Fe_3_O_4_ with a 400 nm particle size was 8 h of reaction time, 60 mL of EG, pH = 8–10, and 1.0 g of PEG. In addition, this study carried out a preliminary study on the preparation of Fe_3_O_4_ using the microwave solvothermal method, which provided a powerful theoretical basis for the preparation of large-size Fe_3_O_4_ using the microwave method.

The magnetic nanocomposite prepared by the optimized Fe_3_O_4_ was used as the chromatographic packing material, and the prepared column had good hydrophobic and stereoselective properties. It also had a good separation effect in the analysis of four sulfonamides.

## Figures and Tables

**Figure 1 nanomaterials-13-01111-f001:**
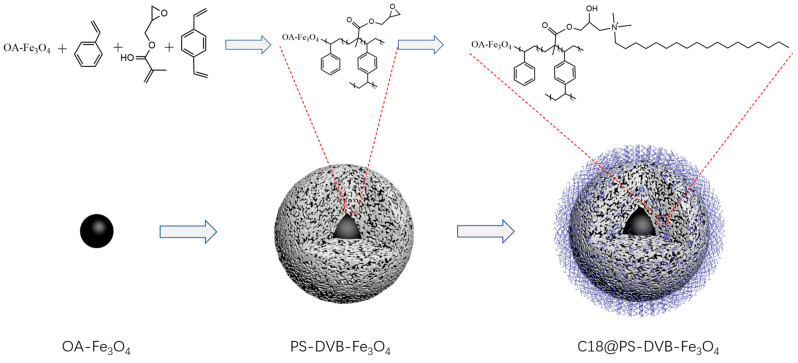
The preparation procedure of the C18@PS-DVB-Fe_3_O_4_ particles.

**Figure 2 nanomaterials-13-01111-f002:**
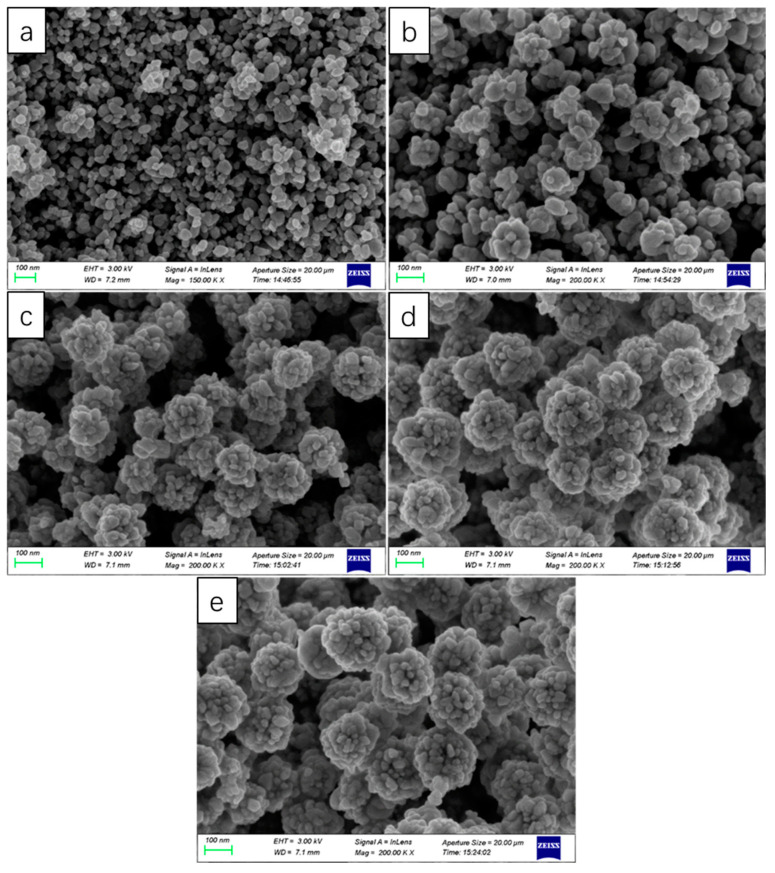
SEM image of Fe_3_O_4_. The amounts were: (**a**) 0.4 g NaOH; (**b**) 0.2 g NaOH; (**c**) 0.1 g NaOH; (**d**) 0.05 g NaOH; and (**e**) without using NaOH.

**Figure 3 nanomaterials-13-01111-f003:**
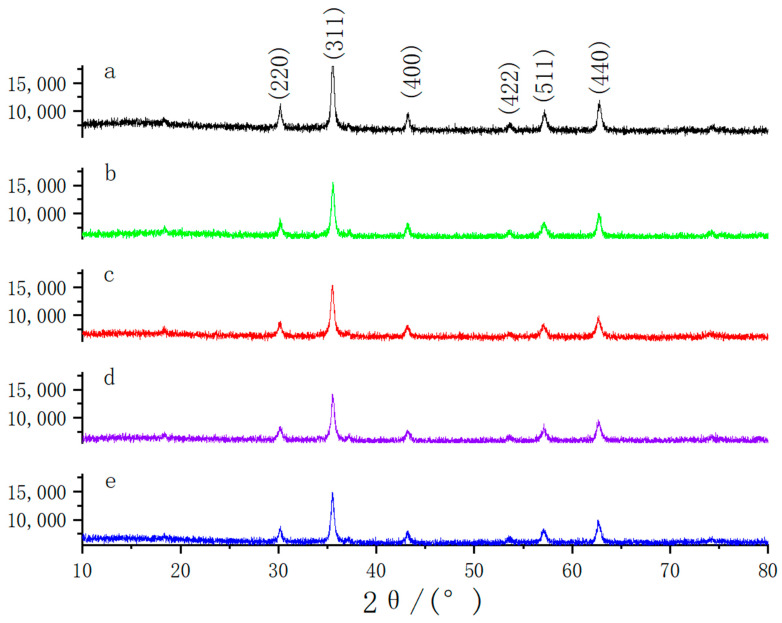
XRD image of Fe_3_O_4_. The amounts were: (**a**) 0.4 g NaOH; (**b**) 0.2 g NaOH; (**c**) 0.1 g NaOH; (**d**) 0.05 g NaOH; and (**e**) without using NaOH.

**Figure 4 nanomaterials-13-01111-f004:**
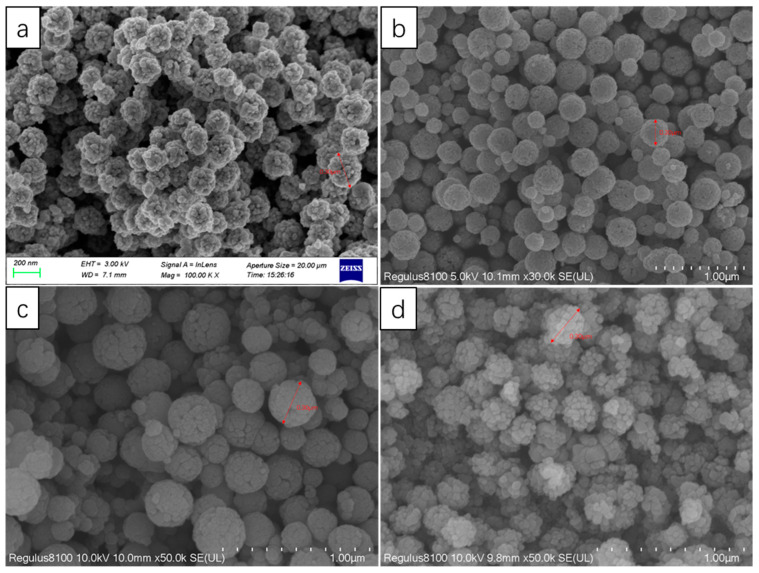
SEM image of Fe_3_O_4_. The times were: (**a**) 6 h; (**b**) 8 h; (**c**) 10 h; and (**d**) 12 h.

**Figure 5 nanomaterials-13-01111-f005:**
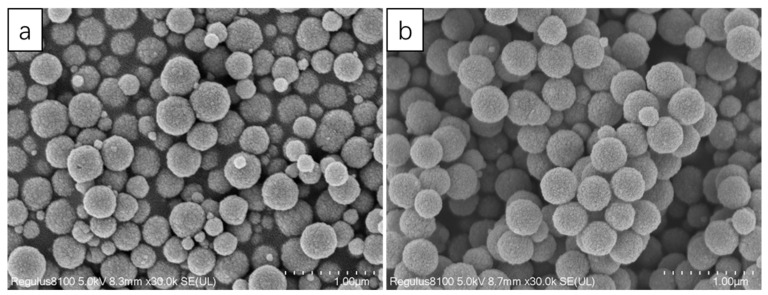
SEM image of Fe_3_O_4_. (**a**) Without using PEG; (**b**) using PEG.

**Figure 6 nanomaterials-13-01111-f006:**
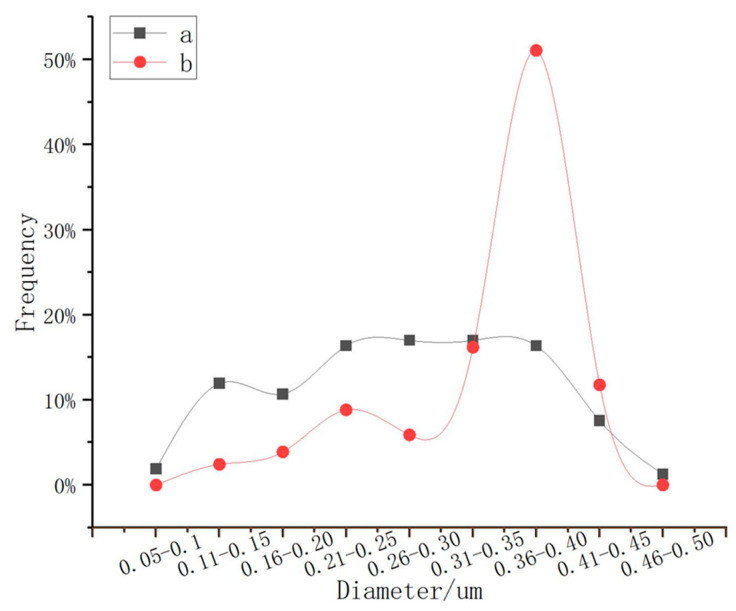
Particle size distribution curves of 150 microspheres. (a) Without using PEG; (b) using PEG.

**Figure 7 nanomaterials-13-01111-f007:**
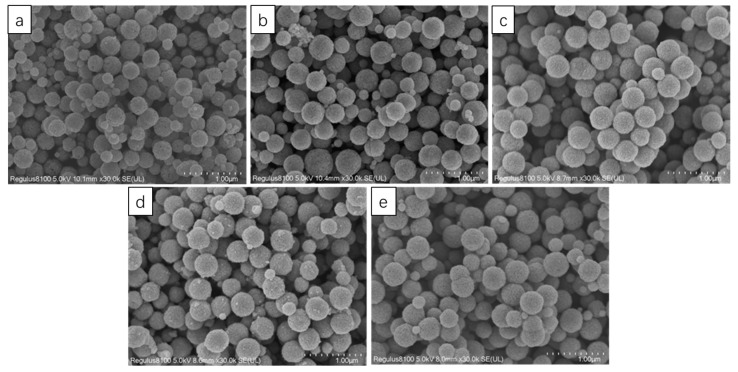
SEM image of Fe_3_O_4_. Solvent amounts: (**a**) 40 mL EG; (**b**) 50 mL EG; (**c**) 60 mL EG; (**d**) 70 mL EG; and (**e**) 80 mL EG.

**Figure 8 nanomaterials-13-01111-f008:**
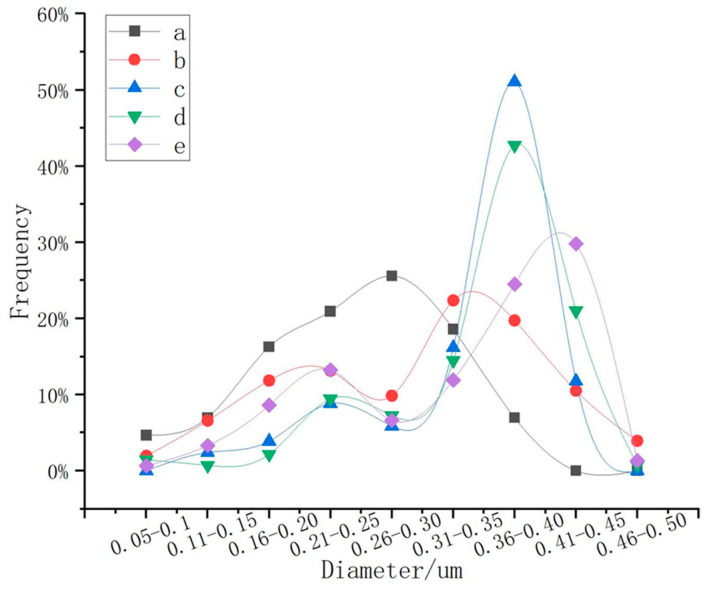
Particle size distribution curves of 150 microspheres. Solvent amounts: (a) 40 mL EG; (b) 50 mL EG; (c) 60 mL EG; (d) 70 mL EG; and (e) 80 mL EG.

**Figure 9 nanomaterials-13-01111-f009:**
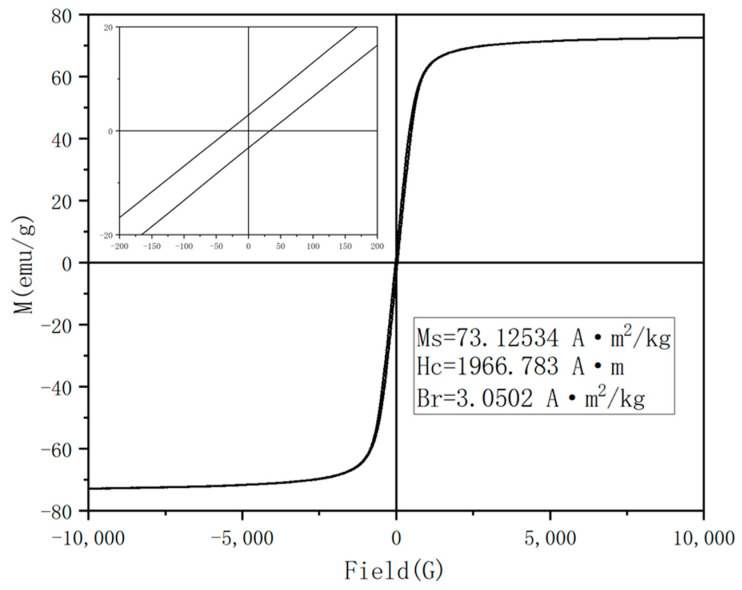
Hysteresis loop diagram of Fe_3_O_4_ nanoparticles.

**Figure 10 nanomaterials-13-01111-f010:**
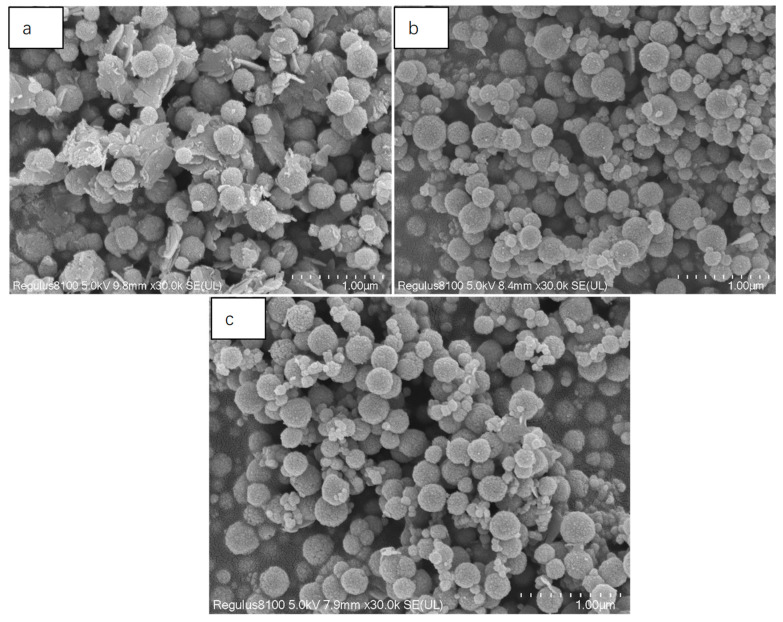
SEM image of Fe_3_O_4_ synthesized by microwave. Reaction time: (**a**) 2 h; (**b**) 3 h; and (**c**) 4 h.

**Figure 11 nanomaterials-13-01111-f011:**
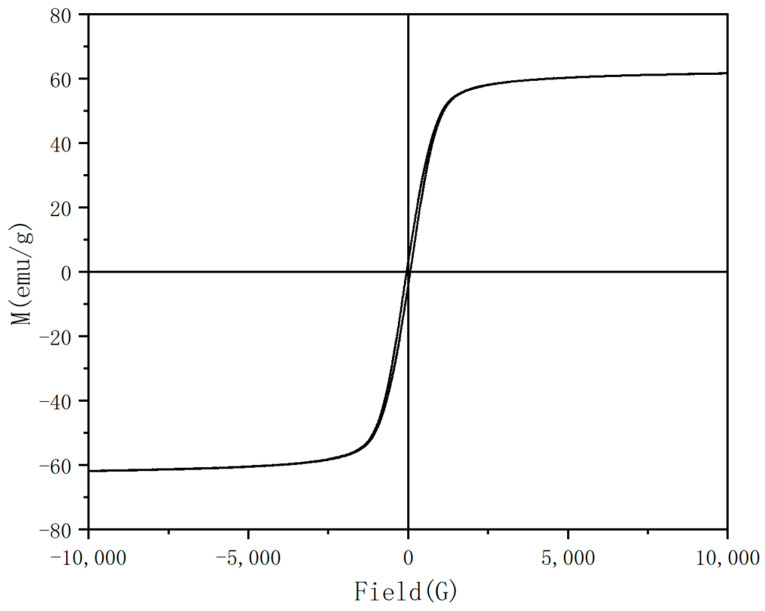
Hysteresis loop diagram of Fe_3_O_4_ nanoparticles by microwave reaction for 4 h.

**Figure 12 nanomaterials-13-01111-f012:**
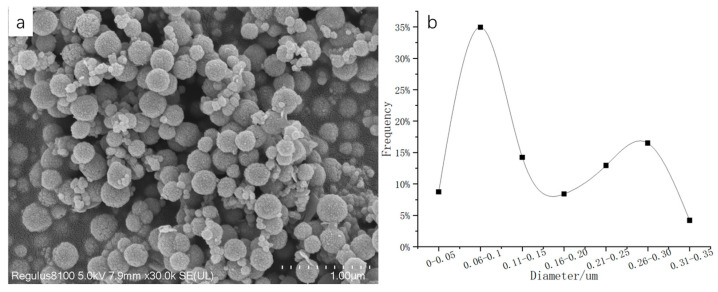
(**a**) SEM image of 3 h preparation of Fe_3_O_4_ by microwave method; (**b**) Particle size distribution curves of 300 microspheres in (**a**).

**Figure 13 nanomaterials-13-01111-f013:**
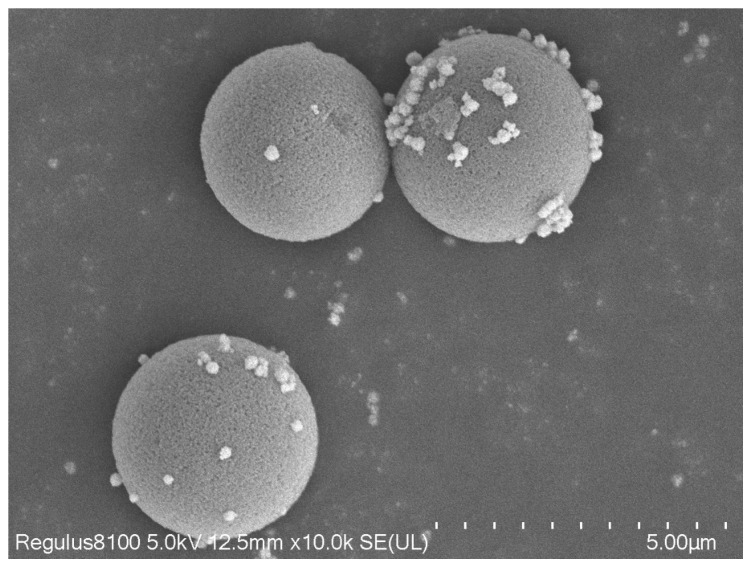
SEM image after organic resin coating.

**Figure 14 nanomaterials-13-01111-f014:**
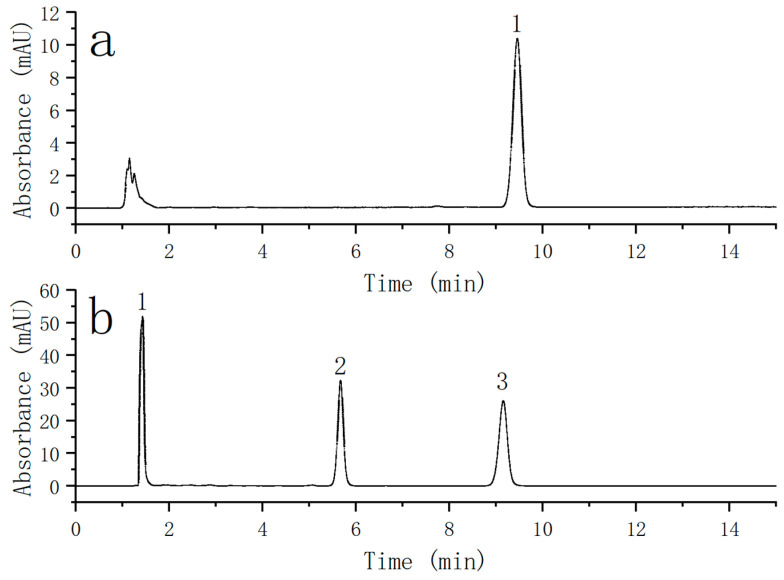
(**a**) Chromatographic separation diagram of naphthalene (1); (**b**) chromatographic separation diagram of uracil (1), ortho-terphenyl (2), and chrysene (3).

**Figure 15 nanomaterials-13-01111-f015:**
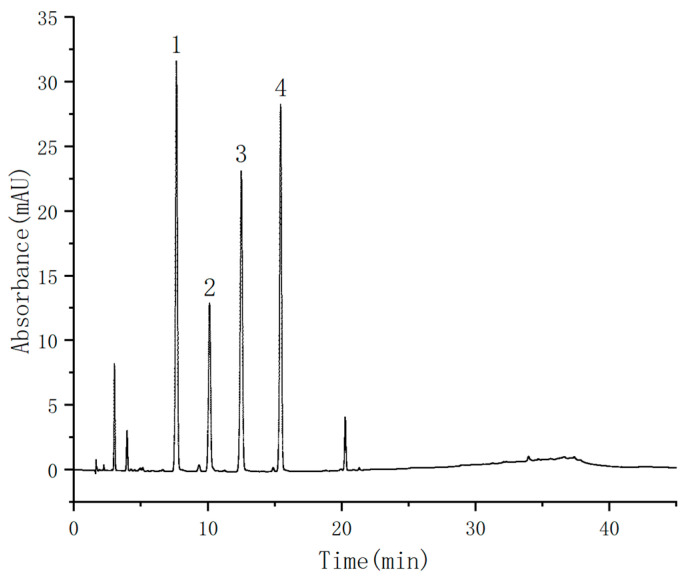
Chromatograms of four sulfonamides: (1) sulfamethyldiazine; (2) sulfamethazine; (3) sulfamethoxypyridazine; and (4) sulfamethoxazole.

**Table 1 nanomaterials-13-01111-t001:** Experimental design parameters.

Serial Number	NaOH/g	Reaction Time/h	PEG/g	EG/mL	Microwave Time/h
A1	0.4	6	0	40	0
A2	0.2	6	0	40	0
A3	0.1	6	0	40	0
A4	0.05	6	0	40	0
B1	0	6	0	40	0
B2	0	8	0	40	0
B3	0	10	0	40	0
B4	0	12	0	40	0
C1	0	8	1.0	50	0
C2	0	8	1.0	60	0
C3	0	8	1.0	70	0
C4	0	8	1.0	80	0
D1	0	8	0	50	0
D2	0	8	0	60	0
D3	0	8	0	70	0
D4	0	8	0	80	0
E1	0	0	1.0	60	2
E2	0	0	1.0	60	3
E3	0	0	1.0	60	4

**Table 2 nanomaterials-13-01111-t002:** Linear equation, detection limit, and relative standard deviation of the four sulfonamides (n = 6).

Target	Linear Range (μg/mL)	Linear Equation	R^2^	RSD%	
				Interday	Intraday
Sulfamethyldiazine	0.1–10	Y = 0.6867X − 2.0467	0.99945	1.6	2.3
Sulfamethazine	0.1–10	Y = 0.5749X − 2.0065	0.99974	3.5	6.1
Sulfamethoxypyridazine	0.1–10	Y = 0.6044X − 0.8220	0.99981	4.3	2.8
Sulfamethoxazole	0.1–10	Y = 0.5871X − 1.1942	0.99923	2.7	5.2

## Data Availability

Not Applicable.

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
