# Peer review of "Preparation and Characterization of Nano-Fe3O4 and Its Application for C18-Functionalized Magnetic Nanomaterials Used as Chromatographic Packing Materials"

_nanomaterials, 2023, doi:10.3390/nano13061111_

Round 1
Reviewer 1 Report
This paper describes the synthesis of Fe3O4 based magnetic particles in very detail. Furthermore, it is interesting that these particles were coated with organic molecules. In this context it is important to note that these are not the first polymer-coated magnetic particles. May be, these materials they were invented more than 30 years ago. In general, I have to say that the description of the experiments is acceptable, whereas the interpretation, in special of the physical properties is missing. Also, the description of the particles is to some extend incomplete; e.g., the particles displayed in Fig. 1 are typical agglomerates. What is the size of the primary particles? A simple analysis of the x-ray diffraction profiles would give a first answer. Also, this method is already more than hundred years old. Apropos, outdated, the dimensions for magnetic properties emu and Gauss are since mor than 50 years replaced by SI units.
Fig. 8 gives a magnetization curve. The authors show e.g. in Figs. 1 and 6 entirely different particles. Do they have the same magnetization curves? I don’t think so. Please show the magnetic data for the different types of particles. Figs. 5 and 7 show particle size spectra. Are these volume fractions, particle number fractions, or what else?
Besides these less appropriate text, also the figures need some improvement. E.g-, The ordinate in Fig. 2 needs improved lettering.
To summarize: Yes, this an interesting topic with a very unique approach. However, this paper is by far not complete, it is a first draft.
Author Response
Dear editor,
We are truly grateful to yours and reviewers’ critical comments and thoughtful suggestions. Based on these comments and suggestions, we have made careful modifications on the original manuscript. All changes made to the text are in red color. We hope the new manuscript will meet your magazine’s standard. Below you will find our point-by-point responses to the reviewers’ comments/questions:
- This paper describes the synthesis of Fe3O4 based magnetic particles in very detail. Furthermore, it is interesting that these particles were coated with organic molecules. In this context it is important to note that these are not the first polymer-coated magnetic particles. May be, these materials they were invented more than 30 years ago. In general, I have to say that the description of the experiments is acceptable, whereas the interpretation, in special of the physical properties is missing. Also, the description of the particles is to some extend incomplete; e.g., the particles displayed in Fig. 1 are typical agglomerates. What is the size of the primary particles? A simple analysis of the x-ray diffraction profiles would give a first answer. Also, this method is already more than hundred years old. Apropos, outdated, the dimensions for magnetic properties emu and Gauss are since mor than 50 years replaced by SI units.
√ Many thanks for your suggestion. In the work, the effect of different NaOH concentration on the particle size of product was investigated deeply, and the results were showed in Fig. 2. When the usage amount of NaOH was 0.4 g, the particle size of magnetic microspheres was the smallest (only about 20 nm), while the particle size of magnetic microspheres was about 200 nm without using NaOH. The magnetic microsphere was formed by the agglomeration of many small particles (about 20 nm). As shown in Fig. 3, the crystalline phases and chemical structures of the prepared particles were determined by XRD. In the XRD diagram, the main peaks of synthesized magnetic microspheres were located at 30.17°, 35.56°, 43.19°, 53.6°, 57.11°, 62.69°, which were agreed with previously reported works about Fe3O4. Therefore, it can be concluded that the prepared particles were magnetic Fe3O4.”
Secondly, with regard to the unit of magnetic properties, emu and Gauss are still commonly used in the direction of materials. For example, emu/g is also used as a unit in the description of magnetic properties in the (Core-shell magnetic Fe3O4@Zn/Co-ZIFs to activate peroxymonosulfate for highly efficient degradation of carbamazepine), (https://doi.org/10.1016/j.apcatb.2020.119136). Thank you very much!
- Fig. 8 gives a magnetization curve. The authors show e.g. in Figs. 1 and 6 entirely different particles. Do they have the same magnetization curves? I don’t think so. Please show the magnetic data for the different types of particles. Figs. 5 and 7 show particle size spectra. Are these volume fractions, particle number fractions, or what else?
√ Many thanks for your suggestion. We are very sorry for the mistake. According to your suggestion, we have sent the different magnetic particles to the third-party testing institutions for obtaining magnetization curves, however, it would take more than a month to obtain the results. We are very sorry for this situation, and we will keep it in mind in our further study.
For the second question, we are very sorry for the incomprehensible presentation, and the “Fig.6 (a) PEG was not added; (b) PEG was added.” has been corrected to “Fig.6 Particle size distribution curves of 150 microspheres. (a) without using PEG; (b) with using PEG.” And the “Fig.8 Particle size distribution curve” has been corrected to “Fig. 8 Particle size distribution curves of 150 microspheres.”
Thank you very much!
- Besides these less appropriate text, also the figures need some improvement. E.g-, The ordinate in Fig. 2 needs improved lettering.
√ Many thanks for your suggestion. According to your suggestion, the ordinate has been corrected as below. Thank you very much!
Fig. 3 XRD image of Fe3O4. (a) 0.4 g NaOH; (b) 0.2 g NaOH; (c) 0.1 g NaOH; (d) 0.05 g NaOH; (e) without using NaOH.
- To summarize: Yes, this an interesting topic with a very unique approach. However, this paper is by far not complete, it is a first draft.
√ Many thanks for your suggestion. We are very sorry for these mistakes. Based on your comments and suggestions, we have made careful modifications on the original manuscript. We hope the new manuscript will meet your expect. Thank you very much!

Reviewer 2 Report
In the present work, the authors studied the synthesis and properties of Fe3O4 particles. The material was prepared by solvothermal and microwave methods. The prepared magnetite was further functionalized by oleic acid coating and used as a filling material in the chromatographic column. A quick separation with a short elution time of different sulfonamides was achieved. The paper is interesting and valuable. It is publishable in Nanomaterials subject to revision.
1.The paper uses a plenty of abbreviations for different chemicals (lines 99-103). It would be helpful to list the chemicals in a table and show their chemical formulas.
2.The XRD peaks should be indexed (Fig. 2). Indicate the Miller indices of the peaks in the figure.
3.SEM images in Fig. 3 should be given on the same scale. Otherwise, it is difficult to compare the particle size.
4.The saturation magnetization of the uncoated particles is remarkably high (Fig. 8). It should be compared with previously studied Fe3O4 particles obtained by different methods (https://doi.org/10.3390/nano11061614, https://doi.org/10.3390/nano12111786).
5.The effect of microwave radiation (section 3.7) should be sufficiently discussed. The results (particle size) should be compared with previous studies of microwave synthesized Fe3O4 nanoparticles.
6.If you studied magnetic properties of coated particles (line 305), please include their magnetization curves in the manuscript.
7.The chromatography efficiency of the C18-functionalized magnetite (retention time) should be compared with other commonly used materials for the separation of sulfonamides.
Author Response
Dear editor,
We are truly grateful to yours and reviewers’ critical comments and thoughtful suggestions. Based on these comments and suggestions, we have made careful modifications on the original manuscript. All changes made to the text are in red color. We hope the new manuscript will meet your magazine’s standard. Below you will find our point-by-point responses to the reviewers’ comments/questions:
- In the present work, the authors studied the synthesis and properties of Fe3O4 particles. The material was prepared by solvothermal and microwave methods. The prepared magnetite was further functionalized by oleic acid coating and used as a filling material in the chromatographic column. A quick separation with a short elution time of different sulfonamides was achieved. The paper is interesting and valuable. It is publishable in Nanomaterials subject to revision.
√ Many thanks for your comments.
- The paper uses a plenty of abbreviations for different chemicals (lines 99-103). It would be helpful to list the chemicals in a table and show their chemical formulas.
√ Many thanks for your suggestion. According to your suggestion, the chemical formulas of main reagents used for the preparation of C18@PS-DVB-Fe3O4 particles have been lised in table S1, however, it would be much more helpful to add the preparation procedure of the C18@PS-DVB-Fe3O4 particles as below. Thank you very much!
Fig. 1 The preparation procedure of the C18@PS-DVB-Fe3O4 particles
Table S1 Chemical structure of reagents used for the preparation of C18@PS-DVB-Fe3O4 particles
Number |
Reagents |
Chemical structures |
1 |
Styrene (ST) |
|
2 |
Oleic acid (OA) |
|
3 |
Divinylbenzene (DVB) |
|
4 |
2,2'-Azobis(2-methylpropionitrile) (AIBN) |
|
5 |
Sodium dodecyl sulfate (SDS) |
|
6 |
Dibutyl phthalate (DBP) |
|
7 |
Benzoyl peroxide (BPO) |
|
8 |
N, N-dimethyloctadecylamine |
|
9 |
Glycidyl methacrylate (GMA) |
- The XRD peaks should be indexed (Fig. 2). Indicate the Miller indices of the peaks in the figure.
√ Many thanks for your suggestion. We have corrected the mistake. Thank you very much!
Fig. 3 XRD image of Fe3O4. (a) 0.4 g NaOH; (b) 0.2 g NaOH; (c) 0.1 g NaOH; (d) 0.05 g NaOH; (e) without using NaOH.
- SEM images in Fig. 3 should be given on the same scale. Otherwise, it is difficult to compare the particle size.
√ Much thank you for your suggestion. We are very sorry for the mistake. According to your suggestion, we have sent different magnetic particles to the third-party testing institutions for obtaining SEM images on the same scale, however, it would take more than a month to obtain the results. In order to make a better comparison, a microsphere with a diameter of 0.3 μm was found in each of the SEM images. We are very sorry for this situation, and we will keep it in mind in our further study.
Fig. 4 SEM image of Fe3O4. (a) 6 h; (b) 8 h; (c) 10 h; (d) 12 h.
- The saturation magnetization of the uncoated particles is remarkably high (Fig. 8). It should be compared with previously studied Fe3O4 particles obtained by different methods (https://doi.org/10.3390/nano11061614, https://doi.org/10.3390/nano12111786).
√ Many thanks for your suggestion. According to your suggestion, Comparisons with other different methods have been added to the manuscript as below, “Compared with the previously reported work on preparation of Fe3O4, the Fe3O4 prepared in this study had much more advantages in particle size control and magnetic properties. For example, the average particle size of the Fe3O4 particles prepared by using chemical precipitation method was 10 nm and the magnetization saturation intensity was 60 emu/g [40]. The similar results can be obtained by using microwave hydrothermal method [41], and the Fe3O4 particles size was about 20 nm and the magnetization saturation intensity was also 60 emu/g. As a result, it could be concluded that the Fe3O4 particles obtained by using solvothermal method possess much higher magnetization saturation intensity and larger partical size.”
- Gerulova, K.; Kucmanova, A.; Sanny, Z.; Garaiova, Z.; Seiler, E.; Caplovicova, M.; Caplovic, L.; Palcut, M. Fe3O4-PEI nanocomposites for magnetic harvesting of chlorella vulgaris, chlorella ellipsoidea, microcystis aeruginosa, and auxenochlorella protothecoides. Nanomaterials (Basel). 2022, 12 (11), 1786.
- Savvidou, M. G.; Dardavila, M. M.; Georgiopoulou, I.; Louli, V.; Stamatis, H.; Kekos, D.; Voutsas, E. Optimization of microalga chlorella vulgaris magnetic harvesting. Nanomaterials. 2021, 11 (6), 1614.
Thank you very much!
- The effect of microwave radiation (section 3.7) should be sufficiently discussed. The results (particle size) should be compared with previous studies of microwave synthesized Fe3O4 nanoparticles.
√ Many thanks for your suggestion. According to your suggestion, the comparison of Fe3O4 nanoparticles prepared in this work with previous studies of microwave synthesized Fe3O4 nanoparticles has been added to the manuscript as below, thank you very much!
“In addition, the particles with reaction time of 4 h were characterized by VSM and the range of particle size distribution was calculated. As shown in Figure 11, the Fe3O4 particles had strong magnetic properties and its magnetization saturation intensity was 62.49 emu/g. and 300 Fe3O4 particles were randomly selected for statistics from Figure 12 (a), and Figure 12 (b) was obtained. It can be concluded that the number of microspheres with a particle size in the range of 60-100nm was the largest, reaching 35%, and the proportion of microspheres with a particle size larger than 200 nm was 34%. Compared with the previous literature, the particle size of Fe3O4 prepared by Savvidou[41] was only 20 nm. The particle size of Fe3O4 prepared by Aivazoglou[42] was 9 nm. It could be seen that the microwave method was mainly used in the preparation of small particle size Fe3O4. This study provides a certain reference significance for the preparation of large size Fe3O4 by microwave.”
Fig.11 Hysteresis loop diagram of Fe3O4 nanoparticles by microwave reaction for
4 h.
Fig.12 Particle size distribution curves of 300 microspheres.
- If you studied magnetic properties of coated particles (line 305), please include their magnetization curves in the manuscript.
√ Many thanks for your suggestion. According to your suggestion, the magnetization curves about microwave synthesized Fe3O4 nanoparticles has been added to the manuscript as below, thank you very much!
Fig.11 Hysteresis loop diagram of Fe3O4 nanoparticles by microwave reaction for 4 h.
- The chromatography efficiency of the C18-functionalized magnetite (retention time) should be compared with other commonly used materials for the separation of sulfonamides.
√ Many thanks for your suggestion. According to your suggestion, Comparisons with other commercial columns have been added to the manuscript as below, “In addition, the comparison of the prepared chromatographic column with Diamonsil C18 (5μm, 250 × 4.6mm) commercial chromatographic column (Fig. S1) was studied, and the results showed that the retention times of four sulfonamides were basically consistent.”
Thank you very much!
Figure S1 Chromatograms of four sulfonamides. (1) Sulfamethyldiazine; (2) Sulfamethazine; (3) Sulfamethoxypyridazine; (4) Sulfamethoxazole. (a) Diamonsil C18 (5μm, 250â…¹4.6mm); (b) Self-made chromatographic column (5μm, 250â…¹4.6mm).

Round 2
Reviewer 2 Report
Authors answered my comments and improved their manuscript. It can be accepted for publication.